# Lactic Acid Bacteria and Yeast Inocula Modulate the Volatile Profile of Spanish-Style Green Table Olive Fermentations

**DOI:** 10.3390/foods8080280

**Published:** 2019-07-24

**Authors:** Antonio Benítez-Cabello, Francisco Rodríguez-Gómez, M. Lourdes Morales, Antonio Garrido-Fernández, Rufino Jiménez-Díaz, Francisco Noé Arroyo-López

**Affiliations:** 1Departamento de Biotecnología de Alimentos, Instituto de la Grasa (CSIC), Ctra. Utrera km 1, Edificio 46, Campus Universitario Pablo de Olavide, 41013 Sevilla, Spain; 2Área de Nutrición y Bromatología, Dpto. Nutrición y Bromatología, Toxicología y Medicina Legal. Facultad de Farmacia, Universidad de Sevilla, C/P. García González, nº2, 41012 Sevilla, Spain

**Keywords:** table olives, starter cultures, GC-MC analysis, volatile composition

## Abstract

In this work, Manzanilla Spanish-style green table olive fermentations were inoculated with *Lactobacillus pentosus* LPG1, *Lactobacillus pentosus* Lp13, *Lactobacillus plantarum* Lpl15, the yeast *Wickerhanomyces anomalus* Y12 and a mixed culture of all them. After fermentation (65 days), their volatile profiles in brines were determined by gas chromatography-mass spectrometry analysis. A total of 131 volatile compounds were found, but only 71 showed statistical differences between at least, two fermentation processes. The major chemical groups were alcohols (32), ketones (14), aldehydes (nine), and volatile phenols (nine). Results showed that inoculation with *Lactobacillus* strains, especially *L. pentosus* Lp13, reduced the formation of volatile compounds. On the contrary, inoculation with *W. anomalus* Y12 increased their concentrations with respect to the spontaneous process, mainly of 1-butanol, 2-phenylethyl acetate, ethanol, and 2-methyl-1-butanol. Furthermore, biplot and biclustering analyses segregated fermentations inoculated with Lp13 and Y12 from the rest of the processes. The use of sequential lactic acid bacteria and yeasts inocula, or their mixture, in Spanish-style green table olive fermentation could be advisable practice for producing differentiated and high-quality products with improved aromatic profile.

## 1. Introduction

Table olives are a fermented vegetable with a pronounced influence on the Mediterranean diet and culture. Nowadays, worldwide production exceeds 2.5 million tons/year [1]. Due to the presence of oleuropein, the fresh fruits are strongly bitter and, therefore, should be appropriately conditioned before consumption. The most common processing styles are: (a) alkali-treated green olives (Spanish-style); (b) ripe olives, obtained by oxidation in an alkaline medium (Californian style); and (c) directly brined olives (Greek style) [2]. 

Lactic acid bacteria (LAB) are the main beneficial microorganisms found in the fermentation of Spanish-style green table olives [3], but yeasts are also always present and provide exciting technological and probiotic features [4]. Together, they form stable biofilms on the olive surface [5,6,7,8] which are ingested by consumers. As a result, the interest in multifunctional starters with adequate technological properties and probiotic potential, as well as the efforts for finding synergy between these two groups of microorganism, has strongly increased. LAB stabilise the product through the production of lactic acid, which lows the pH, whereas the production of enzymes such as lipase and esterase may contribute to the biological hydrolysis of bitter compounds [7,9]. Simultaneously, yeasts improve organoleptic quality [4]. Then, their coexistence provides the table olives with an attractive sensory appeal in which the presence of volatile organic compounds (VOCs) plays an unquestionable role. Such substances are produced, in both the fruit matrix and brine, by the action of endogenous enzymes, such as lipoxygenases, or exogenous, released by the microorganisms. 

There are several studies related to the determination of VOCs in table olives. Sabatini and Marsilio [10] studied the volatile profile in Spanish-style, Greek-style and Castelvetrano-style green olives of the Nocellara del Belice cultivar. The twenty-two VOCs formed during this olive fermentation were significantly affected by the type and time of processing. López-López et al. [11] studied the sensory profile and volatile composition of 24 samples of Spanish-style green table olives, identifying a total of 133 VOCs and finding a trend to separate samples according to sampling time whereas the segregation by olive cultivar was poor. However, none of these studies associated the presence of VOCs with the use of inoculum. 

Recently, De Angelis et al. [12] identified 47 different VOCs during fermentation of directly brined Bella di Cerignola table olives, reporting differences between uninoculated and inoculated treatments with *Lactobacilli* and *Wickerhanomyces anomalus*. Tufariello et al. [13] studied the influence of the type of inoculation on VOCs production in directly black table olives belonging to two Italian (Leccino and Cellina di Nardò) and two Greek (Conservolea and Kalamàta) cultivars, using sequential inoculation of native yeasts and selected LAB starter. De Castro Sánchez et al. [14] reported the influence of the inoculation with *Lactobacillus* on the volatile composition of Manzanilla green olives. Among a considerable number of new VOCs, a remarkable amount of 4-ethylphenol was detected in inoculated olives compared to the uninoculated processes. The same group related the formation of some VOCs with the presence of *Propionibacterium* and *Clostridium* genera in spoilt Spanish-style green table olives [15]. Pino et al. [16] determined the influence of the inoculation with *Lactobacillus plantarum* and *Lactobacillus paracasei* on the VOCs composition of directly brined Sicilian table olives, finding differences between spontaneous and inoculated processes. All these studies show that the addition of starter cultures could have a marked influence on the VOC composition of fermented olives. 

The working hypothesis of this study was to support that the VOCs profile of olive fermentation may be modulated by the addition of starter culture. For this purpose, gas chromatography-mass spectrometry (GC-MS) analysis was used for the analysis of VOCs and diverse multivariate statistical techniques were applied for studying the results.

## 2. Materials and Methods

### 2.1. Olive Fermentations 

Fermentations were carried out in the 2017/2018 season using olives from Manzanilla variety, processed according to the Spanish-style in cylindrical fermentation vessels (9.5 kg olives/5 L liquid). To hydrolyse the oleuropein, fruits were treated with a solution containing 32.4 g/L NaOH, 21.9 g/L NaCl and 0.89 g/L CaCl_2_ (97% purity), for 7 h, until NaOH penetrated 2/3 pulp. To remove the excess of alkali, fruits were washed in tap water for 5 h. Then, olives were placed in fermentation brines containing 120 g/L (w/v) NaCl, 1.3 g/L CaCl_2_ and 0.012 L de HCl. After performing all these operations in the industry, the fermentation vessels were transported to the pilot plant of the Instituto de la Grasa (CSIC, Sevilla, Spain) for their inoculation, fermentation and analysis.

### 2.2. Experimental Design

Two strain of *L. pentosus* (LPG1, Lp13), one of *L. plantarum* (Lpl15) and the yeast strain *W. anomalus* Y12, all of them previously isolated from the biofilm of table olives, were used for single and co-inoculation experiments. Their selection was based on their technological and probiotic properties determined in previous studies [7]. The experimental design consisted of four individual inoculations of each organism (T1, for LPG1; T2, for Lp13; T3, for Lpl15; T4, for Y12), a combination of all them (T5, for Y12+LPG1+Lp13+Lpl15), and a spontaneous process (T6). All experiments were performed in duplicate. 

Previously to the inoculation, LAB strains were grown at 37 °C overnight on Man Rogosa and Sharpe (MRS) broth medium (Oxoid, Basingstoke, Hampshire, England) whereas the yeast was grown on YM broth (Difco, Le Pont de Claix, France) at 28 °C during 48 h. To favour the acclimation of inoculum, culture media were supplemented with 4% NaCl. Previous to inoculation, to remove the medium, cultures were washed and re-suspended in 0.9% sterile saline buffer. Inoculation was executed 1 day after brining for yeasts and at the 9th day for LAB to reach 5 log_10_ CFU/mL and 6 log_10_ CFU/mL in the cover brine, respectively. In the mixed treatment (T5), yeast and LAB were inoculated sequentially after the same periods and population levels, but using 1/3 for each LAB strain in the case of LAB. 

### 2.3. Control Points of Fermentations

At the moment of LAB inoculation (9 days), and at the end of fermentation (65 days), brine were analysed to determine their main physicochemical parameters (pH, salt, free and combined acidity). LAB, yeast and *Enterobacteriaceae* populations were also determined in brine at the end of fermentation and, in the case of LAB, also before inoculation. These parameters were determined according to procedures described in Benítez-Cabello et al. [7]. Rep-PCR with GTG_5_ primer and clustering analysis were used to determine LAB inoculum imposition at 19 days of fermentation when the LAB populations were at the highest level. For this purpose, 10 colonies from each treatment were randomly picked from the highest dilution and their fingerprinting compared with LPG1, Lp13, and Lpl15 profiles according to the protocol described in Benítez-Cabello et al. [7]. Each fermentation vessel was individually analysed.

### 2.4. Olive Brines’ Sequential Extraction and GC-MS Analysis

At the end of fermentation, 100 mL of brines were removed from each treatment and stored at 4 °C until further analysis. The brines’ volatile fraction was submitted to a sequential sorptive extraction with Twisters^®^ (Gerstel, Müllheim an der Ruhr, Germany). The sequential extraction procedure was performed using two polydimethylsiloxane Twisters^®^ in each sample, i.e., first in immersion (SBSE) and then in the headspace (HSSE) [17]. Six mL of the olive brine was placed in a 20 mL vial, and 1.8 gr of NaCl (30%) plus 8 µL of the internal standard 4-methyl-2-pentanol were added (1,044 mg/L final concentration). A special device called Twicester^®^ was used. This device enables to position the Twister magnetically on the wall of a sample vial and, in this way, to keep it immersed and prevent it from brushing against the vial wall. Extraction by immersion was performed for 1 h, and the sample was stirred with a conventional magnetic stir bar (non-coated stir bar) at 200 rpm at room temperature during the extraction process. The headspace extraction was performed by placing a new Twister^®^ in an open glass insert inside the vial and heating the sample in a water bath at 62 °C for 1 hour. In both cases, after extraction, the Twister^®^ was removed with tweezers, rinsed with Milli-Q water, and dried with a lint-free tissue paper. Both Twisters^®^ were then introduced into the same desorption tube and thermally simultaneously desorbed in a gas chromatograph/mass spectrometer (GC-MS). 

Analyses were conducted using an Agilent 6890 GC system coupled up to an Agilent 5975 inert quadrupole mass spectrometer (Agilent, Santa Clara, CA, US) equipped with a Gerstel Thermo Desorption System (TDS2) and a Cooling Injector System CIS-4 PTV inlet (Gerstel, Müllheim an der Ruhr, Germany). The desorption temperature program was the following: the temperature was held at 35 °C for 0.1 min, was ramped at 60 °C /min to 250 °C and held for 5 min. The temperature of the CIS-4 PTV injector, with a Tenax TA inlet liner, was held at −35 °C using liquid nitrogen for the total desorption time and was then raised at 10 °C /s to 260 °C and held for 4 min. The solvent vent mode was used to transfer the sample to the analytical column. A J&W CPWax-57CB column with dimensions 50 m × 0.25 mm and a 0.20 μm film thickness (Agilent, Santa Clara, CA, US) was used, and the carrier gas was He at a flow rate of 1 mL/min. The oven temperature program was the following: the temperature was 35 °C for 4 min and was then raised to 220 °C at 2.5 °C/min (held 15 min). The quadrupole, source and transfer line temperatures were maintained at 150 °C, 230 °C and 280 °C, respectively. The electron ionization mass spectra in the full-scan mode were recorded at 70 eV with the electron energy in the range of 29 to 300 amu.

Compound identification was based on mass spectra matching using the standard NIST 98 library and the linear retention index (LRI) of authentic reference standards. LRIs were calculated by injecting an *n*-alkanes mixture (C_10_–C_40_) under identical conditions as the samples. We considered identified compound the one which mass spectrum and LRI value matched with those of standards, tentatively identified (TI) when mass spectrum matched with those from NIST mass spectral library and LRI value with literature LRI, compound with identification not confirmed when only the mass spectrum of compound matched with those from NIST library, as unknown compounds we include the compounds which mass spectrum reached a low value of probability of right identification in library search report.

### 2.5. Statistical Analyses 

The values of relative peak area of the diverse VOCs found in the treatments were first subjected to analysis of variance (ANOVA) according to treatments. Only those who showed a significant difference between at least two fermentations conditions (Fisher’s LSD post-hoc test) were used later for studying the influence of inoculation. 

The contribution of the diverse inocula to the selected VOCs was also modelled by ANOVA, using treatments as explicative factors and VOCs as dependent variables (tolerance = 0.0001 and confidence interval for *p* = 0.05), with an = 0 constrain (that is, considering T6, the spontaneous process, as a standard or control). The treatments’ contributions were assessed by the corresponding standardised coefficients of the explicative factors. When the contribution was positive, it was estimated that the treatment significantly contributed to the formation of the corresponding VOCs over the levels reached in the spontaneous. On the contrary, treatments with negative coefficients mean that they decreased the presence of the volatile below the level in the spontaneous process (T6).

The relationship between main microbial population or final physicochemical characteristics with the volatile profile was achieved by PLS-R, using a fast algorithm, automatic stop conditions, Jackknife (LOO) validation, as well as centred and reduction of variables. For the study, the final microbial population (LAB and yeast) and the physicochemical characteristics (pH, titratable and combined acidity) were used as independent and the VOCs as dependent variables. To notice that the physicochemical characteristics corresponded to the final conditions when the samples for the volatile profiles were taken, but they did not represent necessarily those in which the compounds were formed, although both may, in some way, summarize the overall fermentation process. The relationships between independent and dependent variables were measured by the respective standardized coefficients of the first for each VOC (the independent variable). Positive (negative) coefficients mean that the independent and dependent variables changed in the same (oppose) direction. 

Besides, the relationships between treatments and volatile profile were also analysed by biplots and bicluster graphs. Biplots are an exciting tool to study simultaneously the relationship between cases and variables since they can represent both (scores and loadings) in the same plot. Both covariance (more appropriate to study the relationships among variables) and form (more useful for segregating cases) biplots were studied. Also, bicluster was suitable for simultaneously clustering observations and VOCs, therefore providing a map of their relationship. 

The statistical analysis was achieved using XLSTAT v2018 (Addinsoft, Paris, France), for ANOVA and PLS analysis, and R package Multbiplot v 2018 [18], for biplot, clustering, and biclustering.

## 3. Results 

### 3.1. Fermentation Process

Table 1 shows the main physicochemical characteristics of the brines at the moment of the LAB inoculation (9 days after olive brining). LAB and yeast strains were inoculated in a pH ranging from 6.19 to 6.33, titratable acidity of 0.09–0.14%, combined acidity of 0.12–0.15 Eq/L, and a salt concentration of 6.61–6.77%. At the 19th day of fermentation, LAB inoculum imposition was determined by molecular methods, finding that the frequency of isolation of the strains LPG1, Lp13 and Lpl15 in their inoculated treatments (T1, T2, and T3, respectively) were 100%. However, Lp13 also was detected (100% frequency) in the rest of the treatments (T4, T5, and T6), showing that this strain has a high ability for brine colonization. All treatments developed safe final pH values (<4.5). However, T4, inoculated only with the yeast Y12, had a particular performance since its pH was higher than the values observed for the rest of the inocula, although the final difference was only significant regarding T1. However, its titratable acidity was significantly lower than the other treatments. On the contrary, treatment inoculated with LPG1 was the most technologically efficient, reaching the significantly lowest pH value (see Table 1). NaCl concentration was similar in all treatments (7.47 ± 0.21% average).

Regarding LAB growth, they were not detected in any treatment before inoculation. At the VOCs sampling time, they have reached similar populations in all inoculated treatments, although their average value (6.94 log_10_ CFU/ml) in the spontaneous fermentation was the highest at the end of fermentation (65 days). On the contrary, no significant differences between treatments were found in the yeast population at the end of the process (5.99 ± 0.10 log_10_ CFU/mL average value). *Enterobacteriaceae* were never detected during the process. 

### 3.2. ANOVA Analysis

A total of 131 VOCs were determined in the brines from the 12 fermentation trials (6 treatments in duplicate). Results were expressed as relative area values respect to the internal standard (see Appendix A). The chemical group with the highest number of compounds was alcohols (32) followed by ketones (14), aldehydes (9) and volatile phenols (9). A first ANOVA screening of the VOCs according to treatments (Appendix A) showed that the levels of only 71 compounds were significantly different between at least two fermentation processes. Therefore, 60 VOCs were produced regardless of the process and represent a common profile which, at least in the current fermentation conditions, could always be found and included both identified and not assigned formula components. Because in this study the interest was focused exclusively on those which presence could be attributed to the inocula, the compounds not significantly different among treatments were not considered for further analysis.

To investigate the relationships between the inoculated starter cultures and the initially significant VOCs, the ANOVA was again repeated with only these response variables, but using T6, the spontaneous process, as reference. As a result, the contribution of each treatment to the concentration of each volatile (versus the spontaneous) was evaluated through the standard coefficient of the respective models. Due to the large number of VOCs remaining in the study, only a few examples of treatments contributions to the formation of volatile will be illustrated graphically (Figure 1) while the rest are summarised (including only significant coefficients) in Table 2.

In the case of 2-phenylethyl acetate (Figure 1A), the treatments 4 and 5 promoted the formation of this compound due to the presence of the yeast in the inoculum. Interestingly, this compound was stimulated not only by the presence of the yeast but also LPG1 led to an important contribution and, therefore, the presence of this LAB did not interfere with its possible formation but even stimulated it. However, this effect was not observed in the treatments inoculated with Lp13 or Lpl15, since their 2-phenylethyl acetate contents were similar to those in the spontaneous process. Hence, the behaviour of LPG1 differs from Lp13 and Lp115 regarding the formation of 2-phenylethyl acetate. Similarly, 3-methyl-1-butanol was promoted by the presence of the yeast (Figure 1B); however, Lp13 had a marked negative effect on its formation, which is also reflected in T5, where the joint presence of Lp13 and Y12 has practically prevented its presence. The effects of inoculation treatments on these two volatile substances are reflected in Table 2 with a sign (+ means promotion or increase versus the spontaneous process whereas - indicates prevention or decrease) and the coefficient value (the large the value the most important the effect while the absence of data means not significant contribution). A similar methodology was also followed for the other 69 compounds. It should be noted that due to their standardization, the contributions are independent of the volatile concentrations. Besides, due to the large number of compounds, only an overview of the inoculation with the diverse LAB, yeast, and their mixtures can be commented on. For a detailed relationship for specific compounds, please see Table 2. Overall, inoculation with LGP1 (T1) reduced (negative sign) the production of several VOCs with respect to the control (T6, spontaneous fermentation), with methanol, β-damascenone, and other unknown volatiles among them. On the contrary, it promoted (by itself or by allowing its formation by the spontaneous yeasts, by chemical reaction, or a combination of transformations pathways) of many others like 2-phenylethyl acetate, 2-butanol, 1-butanol, 3-methyl-3-buten-1-ol, *cis*-2-penten-1-ol or 2-methyl-3-hexanol. Therefore, the analysis of VOCs was useful to study the influence of LPG1 presence on, at least, an aspect (VOCs) of the metabolomic related to the fermentation process. 

Particularly interesting was the effect of the inoculation with Lp13 strain. In this case, almost all standardized coefficients had a negative sign (only that for 1–butanol was positive); that is, its presence had an important effect on the volatile composition reduction. On the contrary, the inoculation with the Lpl15 had an almost neutral effect on the formation of the VOCs since the significant coefficients were very reduced and had both positive and negative signs; however, it promoted the formation of methanol, isoxylaldehyde and 4-ethylphenol, while reducing that of coumarin, 5-*tert*-butylpyrogalol and vanillin. 

A radically opposed behaviour was shown by yeast inoculated treatments. The inoculation with Y12 was determinant for increasing dramatically the concentration of most of the VOCs over the spontaneous process (T6) with only a few coefficients with a negative sign. Among the compounds which formation promoted inoculation with Y12 were 1-butanol, ethanol, methyl acetate, ethyl acetate, 2-phenylethyl acetate, or 2-methyl-1-butanol, to mention only a few of them; but, it depressed the levels of methanol, coumarin, and vanillin. Therefore, it was evident that the inoculation with only the yeast increased the amount of VOCs of the fermented olives. In most cases, this increase was inversely related to the production of free acidity, combined acidity, and the subsequent high pH (Table 2, PLS regression). This inverse relationship shows a competence between the productions of one or several compounds *vs* the others. Finally, when the yeast was inoculated together with the rest of LAB strains (T5), the volatile compounds content was more abundant than in the case of the spontaneous treatment, although some of the compounds found in the presence of the yeast like ethanol, 1–heptanol, or *cis*-5-octen-1-ol were reduced with respect to T4 and remain similar to T6 (spontaneous), possibly due to the competence of the LAB also present in T5. The effect on some VOCs like 2-methyl-1-butanol, 3-methyl-1-butanol, 1–heptanol, or *cis*-5-octen-1-ol could be directly associated with Lp13 presence, which did not promote their production. Therefore, the use of the only LAB in the starter cultures decreased or not affected the production of VOCs while the inoculation of only Y12 increased them. However, when both groups of microorganisms were mixed (T5), the yeast metabolites were affected (Table 2), revealing a competence between both groups. 

### 3.3. PLS Analysis

The overall PLS-R model quality (one component) was reduced since Q^2^cum explained low variances of both independent (Q^2^X = 0.404) and dependent variables (Q^2^Y = 0.305), although it may also be due, at least partially, to the noise introduced when working with numerous non-significant variables. Overall, the most influential variables in the model (Figure 2A) were titratable acidity, pH, and combined acidity (which in table olives are always strongly related to the first two) while LAB and yeasts counts were never significant, although the relationships could only be established with a reduced number of VOCs. An example of the coefficients is shown in Figure 2B, which corresponds to ethanol. The negative sign for titratable and combined acidities mean that high production of them (and their associated low pH), lead to low ethanol production (lactic acid fermentation predominated over yeast fermentation and in some way reduced the ethanol production). This opposed trend between these physicochemical characteristics and volatile composition was common to most compounds but, especially, to alcohols (Table 2). However, numerous compounds were also unaffected (2-phenylethyl acetate, 3-methylbutanoic acid or dimethyl sulfoxide), indicating a possible compatible, metabolic pathway or absence of competence for the nutrient/substrates between both LAB and yeasts (Table 2), with several of them being related to alcohols as well (e.g. 1-hexanol or *cis*-3-hexen-1-ol). Only in the case of vanillin, the production of lactic acid did not lead to a decrease in its formation. 

### 3.4. Biplots Analysis

The analysis showed that two or three Factors accounted for 64.1 and 75.4% of the variance, respectively. Most of the treatments (cases) were well represented (big size names) onto the first two factors plane (Figure 3A) while T1 and T6 treatments were better associated, at least partially, to F2 or F3 (Figure 3B). Besides, the study showed that four clusters produced the best segregation among fermentations. In this case, the boundaries delimited by the Voronoi lines help to recognize the appropriate influential areas and to ascribe VOCs to fermentation clusters (Figure 3).

As shown by the big sizes of their names, T2, T3 (one replicate), T4 and T5 are well represented on the F1/F2 plane. Similarly, variables with large arrows are better represented than those with the shortest lengths. The plot shows that T2 treatment (inoculated with Lp13) and T4 (Y12) represent two very different fermentation volatile profiles with T2 being characterized by a scarce volatile content (methyl lactate (AK in the graph), and unknowns N (BL) and A (AZ), in slightly lower proportions) while T4 was abundant in many of them (4-methylguaiacol (AO)), 1-heptanol (T), unknowns B (BA), D (BB), F (BE), Q (BO), E (BD), 2-methyl-1-butanol (K), 3-methyl-1-butanol (L), ethyl 5,6-dimethyl nicotinate (AD), 6-hepten-1-ol (U), *cis*-5-octen-1-ol (V), ethanol (G), ethyl acetate (B), 1-pentanol (N), benzyl alcohol (W), 2-phenylethyl acetate (D), and 2-phenylethanol (X)). The other two treatments show intermediate values of these two volatile profiles plus some representative compounds of their fermentations. Thus, T3 and T6 (in lower extension) were also characterized by unknown U (BR) and P (BN), purpurocatechol (AI), furfuryl methyl ether (AF), α-isophorone (AW), unknown M (BK), isovanillic acid (AR), or methoxyeugenol (AU). Besides, T5 (which in the ANOVA was also identified with abundant VOCs and it is also at a large distance from the origin) have a reduced number of characteristic compounds as would correspond to treatments inoculated with the mixture of microorganisms. In this case, only 3-methylbutanoic acid (E) could be the most representative while also may participate other VOCs like 2-butanol (H), *cis*-3-hexen-1-ol (R), 3-methyl-3-buten-1-ol (M), *cis*-2-penten-1-ol (O), 2-methyl-2-buten-1-ol (P), or 1-hexanol (Q) which are included in its Voronoi area. On the contrary, T1 (inoculated with LPG1) was not well represented and will be commented later. 

In the plane F1/F3, T1 (one replicate), T5, T6 or T3 had very poor contributions. However, in F2/F3 (Figure 3B), T5 and T1 (one duplicate) were well represented, but the other replicate of T1 was still close to the centre, indicating that, overall, this replicate had a low representation. A similar situation was also observed for one replicate of T6 and T3. Therefore, in the F2/F3 plane, the volatile compounds best related to T3 and T6 (in lower proportion) were: 3-ethylpyridine (AN), methoxyeugenol (AU), 4-ethylguiacol (AP), furfuryl methyl ether (AF), unknowns G (BF), P (BN), M (BK), K (BI), O (BM), and α-isophorone (AW); however, as T3 is closely related to T6 this means that the inoculation with Lpl15 produce, in general, quite similar volatile compounds than T6 (spontaneous process). Besides, T1 and T5 may be related to 3-methylbutanoic acid (E), α-terpineol (AX), and geraniol (AY), with the last two compounds being better represented in this plane than in that of F1/F2, where their contributions were markedly lower. 

Therefore, overall, the biplot showed that most of the VOCs were associated with the F1 axis (most positively and a few of them negatively). Therefore, this axis was the most influential for the treatment segregation, particularly between T4 (rich in many volatile compounds, inoculated with Y12) and T2 (Lp13, abundant in only a few components) while the rest of the treatments were more similar, particularly T1 (inoculated with LPG1) and T5 (with mixture of LAB and Y12), and limited regarding their contributions to volatile compounds. The reduction of volatile composition in T5 could have been caused by Lp13 who, in the ANOVA table, showed a clear negative effect on the formation of volatile compounds. On the contrary, the F2 axis was associated with a reduced number of VOCs both positively (3-methylbutanoic acid (E), α-terpineol (AX) and geraniol (AY)) and negatively (furfuryl methyl ether (AF), α-isophorone (AW), unknowns P (BN), G (BF), isovanillic acid (AR), or methoxyeugenol (AU)), which were linked to T5 and T1 and, T3 and T6, respectively. Therefore, F2 axis was also efficient for segregating these two groups, although one should always have in mind that, in half of the T1, T3 and T6 replicates, the presence of the volatile compound was not particularly relevant.

Regarding relationships among variables, they could be deduced from the angles of their respective arrows. It is evident that those pointing to the right in Figure 3A are related, and their production may be assigned to Y12. On the contrary, those looking towards the left (methyl lactate (AK), unknowns N (BL), U (BR), or purpurocatechol (AI) are strongly related among them and possibly linked to the metabolic pathways of the strain Lp13. Non-related to these VOCs might be those variables pointing up in the plot (Figure 3A) like 3-methylbutanoic acid (E)) and down as furfuryl methyl ether (AF), α-isophorone (AW), unknowns P (BN), G (BF), isovanillic acid (AR), methoxyeugenol (AU), or 6-methyl-3,5-heptadien-2-one (AH), with these two groups showing, in turn, opposed relationship among them. Besides, strong relationships may be observed in Figure 3B for volatile compounds pointing to the left (associated with T3 and T6) and right (linked to T5 and T1), but opposed between them.

### 3.5. Biclustering Analysis 

An appropriate presentation of the whole relationship between VOCs and treatments may also be achieved by biclustering; that is, according to treatments and VOCs simultaneously (Figure 4). The clustering of treatments also led to four clusters (indicated as b1-4) while the VOCs were grouped into four other big clusters (v1-4). The first (b1) was composed of only T2 treatment (inoculated with Lp13); the second (b2) consisted of T3 (Lpl15) and T6 (spontaneous); the third (b3) included T1 (LPG1) and T5 (mixture of Y12 and LAB strains), with a possible segregation between them; and the forth (b4) was devoted to only T4 (Y12). Therefore, this segregation was similar to that achieved in the biplot analysis where only Lp13 (T2) and Y12 (T4) led to individually differentiated VCOs profiles. Combining this segregation with the volatile compounds, it may be observed that the profile of T4 (Y12) was characterized by the high production of compounds such as methyl 4 (methylamino) benzoate (AM), ethyl 5,6-dimethylnicotinate (AD), unknowns B (BA), C (BB), D (BC), E (BD), F (BE), G (BF), H (BG), W (BS), Q (BO), and S (BP), ethanol (G), 6-hepten-1-ol (U), 2-methyl-1-butanol (K), 3-methyl-1-butanol (L), β-damascenone (AB), *cis*-3-hexenyl acetate (C), 5-tert-butylpyrogallol (AT), *cis*-5-octen-1-ol (V), 4-methylguaiacol (AO), 1-heptanol (T), 1-butanol (J), methyl acetate (A), 2-methyl-3-hexanol (S), 2-phenylethyl acetate (D), benzyl alcohol (W), 1-pentanol (N), ethyl lactate (AC), 4-ethylphenol (AP), 2-phenylethanol (X), 1-hexanol (Q), *cis*-2-penten-1-ol (O), 3-methyl-3-buten-1-ol (M), *cis*-3-hexen-1-ol (R), 2-methyl-2-buten-1-ol (P), 3-ethylpyridine (AN), methoxyeugenol (AU), isovanillic acid (AR), 6-methyl-3,5-heptadien-2-one (AH), and iridomyrmecine (AJ). On the opposite side is the T2 treatment (inoculated with Lp13), which is low or minimal in most of the compounds but high in dimethyl sulfoxide (AA), vanillin (AV), unknown A (AZ), N (BL), methyl lactate (AK), and 2-ethenyl-2-butenal (Y). The cluster consisting of T6 and T3 treatments is low or minimal in VOCs clustered in v1 while high or moderated in those included in v2. A more detailed specific relationship may be read directly from the graph (Figure 4). 

## 4. Discussion

In this work, a total of 131 VOCs, formed during Spanish-style table olive fermentation inoculated with diverse LAB and yeast native strains, have been determined using GC-MS analysis. Panagou and Tassou [19] studied through GC analysis the VOCs during the fermentation of Conservolea variety green olives inoculated with *L. pentosus* and *L. plantarum* strains, finding that ethanol, methanol, acetaldehyde, ethyl acetate, and isobutyric acid were the major VOCs identified during fermentation, some of them also found in this work. Recently, Cosmai et al. [20] applied SMPE/GC-MS analysis to study the VOCs of directly brined green table olives from *Bella di Cerignola variety* in treatments inoculated with *W. anomalus* and strains of *L. pentosus* and *L. plantarum*. They specially reported higher levels of lipoxygenase pathway-derived compounds as 1-hexanol or *cis*-3–hexen-1-ol in treatments inoculated with the yeast in which these compounds were overrepresented in treatments inoculated with *W. anomalus* Y12. In this paper, similar results were obtained for the last compound. Tufariello et al. [13] reported that the use of sequential inoculation of yeast and *Lactobacilli* species in directly brined olives affected VOCs. Alcohol and ester contents increased during starter-driven fermentations, but with significant differences among olive cultivars, and always in higher concentrations than in the corresponding spontaneous fermentations. No variation of hydrocarbons and terpenes was detected between spontaneous and starter-driven fermentations. 

One of the most desirable objectives of designing an inoculum is to improve the organoleptic profile of olive fermentations, especially aroma [21]. In this work, very relevant differences between the VOC levels in the brine obtained after fermentation processes appear to depend on the microorganism used as inoculum, especially when yeasts are involved in the fermentation process. Hence, Sabatini et al. [22] observed that ethanol was produced in brine medium mostly by yeasts fermentation (alcoholic fermentation) and, in a lesser extent, during lactic acid fermentation (heterolactic fermentation). Our results are in agreement with them, and ethanol was produced mainly in the brine inoculated with yeast, doubling the amount found in fermentation processes carried out by *Lactobacillus* strains. Similar results were found for other alcohols closely related to alcoholic fermentation pathways such as isoamyl alcohols (2-methyl and 3-methyl-1-butanol), or 1-butanol. Sabatini and Marsilio [10] also detected by GC/MS analysis diverse VOCs, comprising alcohols, aldehydes, ketones, esters as well as acids, formed during olive fermentation of Spanish-style, Greek-style and Castelvetrano-style green olives of the Nocellara del Belice cultivar. Their results suggested that different processing technologies significantly affected the VOCs of samples, as well as the time of processing. Recently, Pino et al. [16] using GC-MS analysis found that the addition of *L. plantarum* and *L. paracasei* as starters significantly modified the volatile profile of directly brined Sicilian table olive fermentations. Specifically, compounds responsible for fruity and floral notes, such as methyl 2-methylbutanoate and phenylethyl alcohol, highly increased, while isoamyl alcohol and ethanol decreased compared to non-inoculated samples. The high content of alcohols in un-inoculated brine samples could be related to yeast metabolic activities, which was mainly dominated by *W. anomalus*, but this yeast species were also present during LAB inoculated fermentations. 

Acetic acid esters are compounds formed by condensation between acetic acid and an alcohol. *W. anomalus* yeast has been reported to be an acetic acid ester producer [23]. The significant different high contents of ethyl acetate and 2-phenylethanol acetate are another relevant result of this work. This fact showed that, in table olive fermentation, this yeast might also develop its capability to produce such kinds of esters. 

4-Ethylphenol, a compound with an unpleasant aroma, could be produced during lactic acid fermentation [24,25]. On the one hand, Randazzo et al. [26] studied the VOCs produced by different *Lactobacillus* strain inocula in brines of Nocellara Etnea table olive fermentations. Among strains compared, they studied the effect of a pure culture of one *L. plantarum* strain and other *L. pentosus* strain. They did not find a significant difference concerning the production of 4-ethylphenol. However, de Castro et al. [14] suggested that 4-ethylphenol formation is strain-dependent. Our results suggest that *L. plantarum* Lpl15 strain has a high capability for the production of this volatile phenol and support the idea of strain-dependent production. 

## 5. Conclusions

The statistical approach used in the present work has allowed identifying the main modification in the volatile profile produced by inoculation with diverse starter cultures. Our study has demonstrated that the type of inoculum modulates the volatile composition of the final product significantly. The inclusion of yeast in the inoculum increases the production of VOCs while the presence of *Lactobacillus* alone, in general, decrease the concentrations of some compounds or keep them at the same levels than in the spontaneous process. This lack of impact on the VOCs by *Lactobacillus* strains may be explained because the emphasis when selecting starters was mainly focused on the acidification and the pH lowering characteristics. However, as the process is better known from the microbiological point of view, the introduction of genomic methodologies and the application of more accurate and sophisticated methods for the identification of metabolites formed during the process could make possible the design of inocula with wider and better-identified characteristics, including their aromatic profile. Therefore, to enhance the organoleptic characteristics of final products, the inclusion of yeasts in the inoculum appears as a promising alternative. By studying in detail, the relationships between the VOCs formed and the sensory characteristics, appropriate selection of yeast could be achieved. Besides, the relationships found in this work between starter cultures and VOCs may facilitate further studies on the numerous metabolic transformation occurring in table olive fermentations.

## Figures and Tables

**Figure 1 foods-08-00280-f001:**
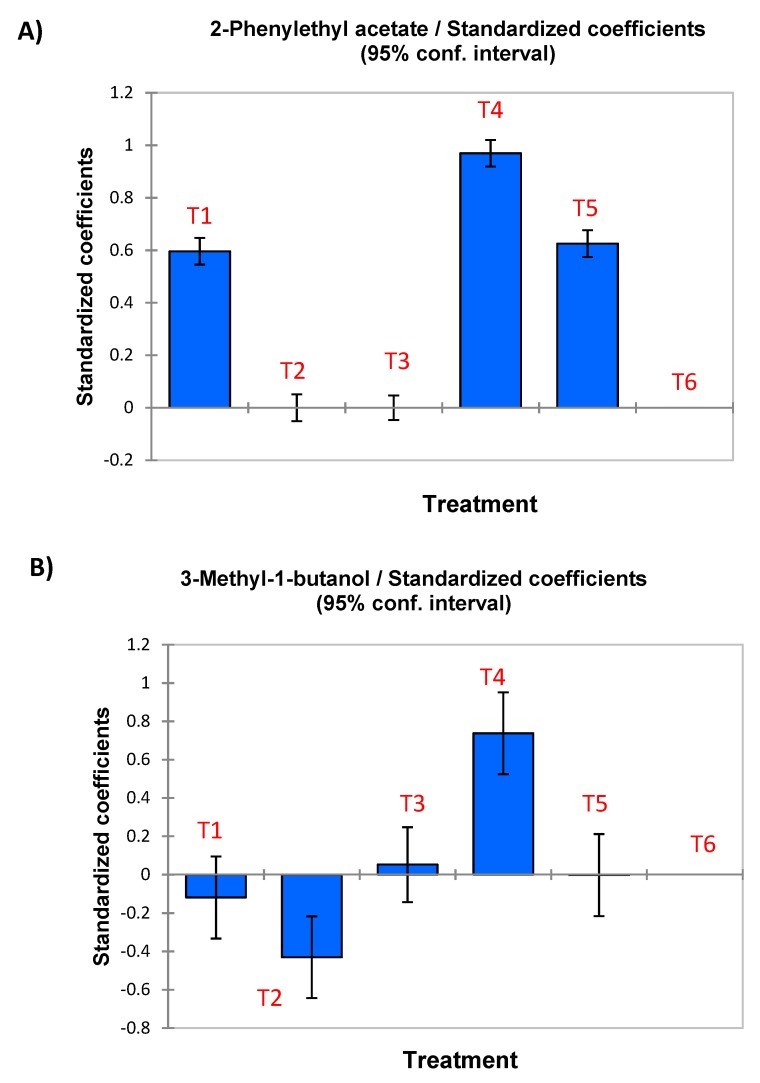
Standardised coefficients obtained after the ANOVA analysis (a_n_ = O, equivalent to stablish T6 treatment, spontaneous fermentation, as standard) for two selected VOCs: (**A**) 2-phenylethyl acetate, and (**B**) 3-methyl-1-butanol.

**Figure 2 foods-08-00280-f002:**
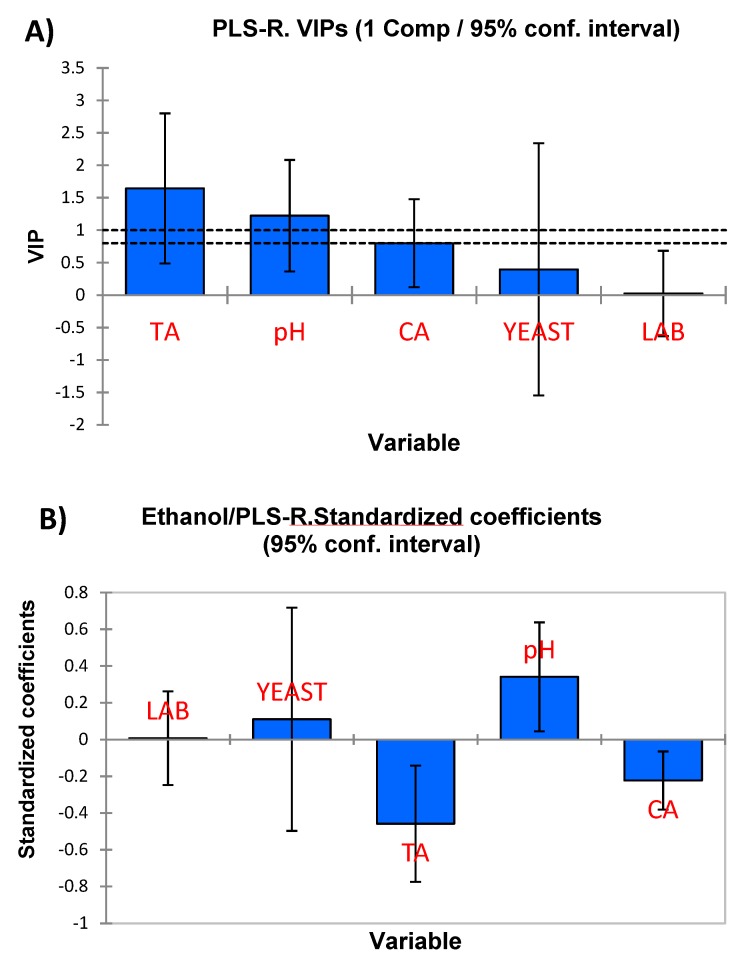
PLS-R analysis, using final physicochemical and microbiological characteristics as independent variables and VOCs as dependent. (**A**) Variable importance for the projection and (**B**) Significant standardised coefficients for the presence of ethanol.

**Figure 3 foods-08-00280-f003:**
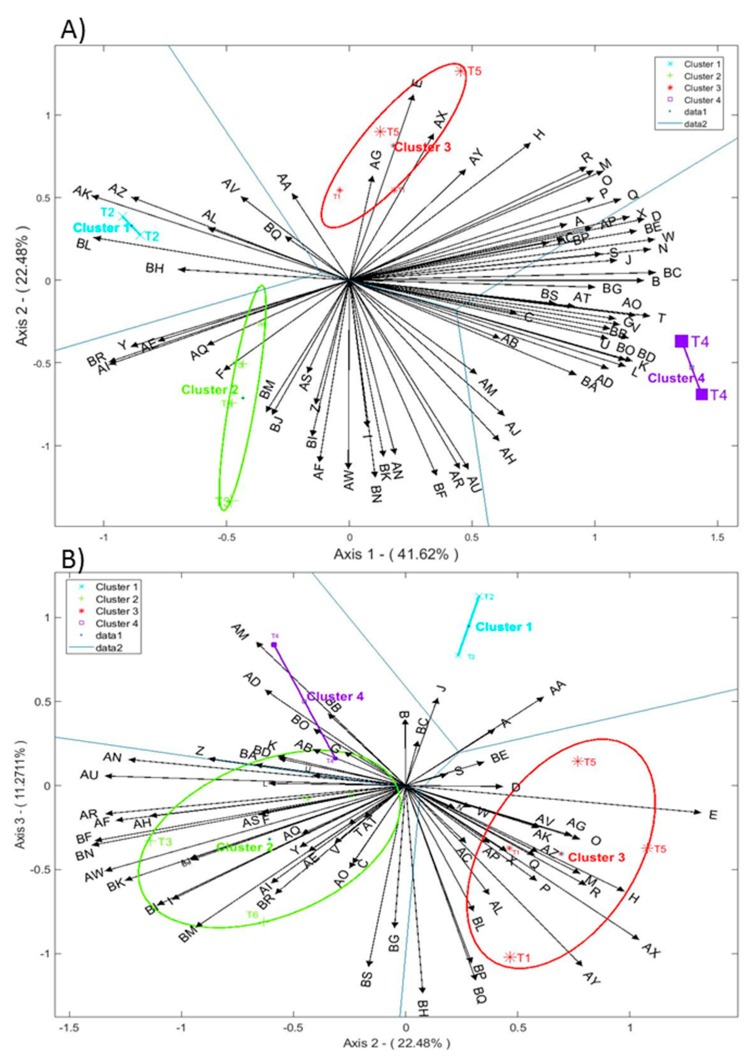
Biplot, including clustering and Voronoi borders, representing the projections of cases scores and variables loadings onto axis F1 vs F2 (**A**) and F2 vs F3 (**B**). Contributions of cases are proportional to the size of symbols and letters (see Table 2 for the meaning of symbols) while those of VOCs (see Table 2 for codes) are proportional to the length of their arrows.

**Figure 4 foods-08-00280-f004:**
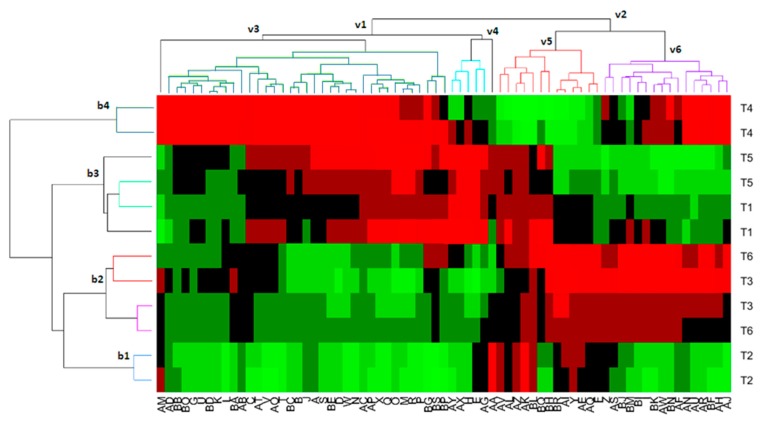
Bicluster plot of the relationship among treatments and VOCs. The presence of volatile is proportional to the colour scale, ranging from red (major) to green (low). See Table 2 for the meaning of the symbols for treatments and the codes of the VOCs. b1-b4 and v1-v4 refer to cluster from treatments and VOCs, respectively.

**Table 1 foods-08-00280-t001:** Microbiological and physicochemical analysis of the treatments at the moment of LAB inoculation (I, 9 days) and at the end of fermentation (F, 65 days). Inoculum imposition was determined by rep-PCR with GTG_5_ primer at 19 days of fermentation.

Treatment	Inoculum Strain	Inoculum Imposition	TA-I * (%)	TA-F (%)	pH-I	pH-F	CA-I ** (Eq/L)	CA-F (Eq/L)	Salt-I (%)	Salt-F (%)	LAB-F (Log_10_ cfu/mL)	Yeasts-F (Log_10_ cfu/mL)	*Enterobacteriaceae*-F (Log_10_ cfu/mL)
**T1**	LPG1	100%-LPG1	0.14 (±0.06) ^a^	0.77 (±0.03) ^a^	6.29 (±0.06) ^a^	4.06 (±0.11) ^a^	0.15 (±0.00) ^a^	0.17 (±0.02) ^a^	6.67 (±0.02) ^a^	7.60 (±0.25) ^a^	5.97 (±0.02) ^a^	5.95 (±0.08) ^a^	Nd
**T2**	Lp13	100%-Lp13	0.09 (±0.01) ^a^	0.79 (±0.00) ^a^	6.31 (±0.11) ^a^	4.18 (±0.00) ^a,b^	0.15 (±0.00) ^a^	0.15 (0.00) ^a^	6.44 (±0.37) ^a^	7.30 (±0.09) ^a^	5.98 (±0.25) ^a^	5.93 (±0.10) ^a^	Nd
**T3**	Lpl15	100%-Lpl15	0.18 (±0.11) ^a^	0.78 (±0.04) ^a^	6.19 (±0.16) ^a^	4.21 (±0.02) ^a,b^	0.15 (±0.01) ^a^	0.16 (±0.01) ^a^	6.61 (±0.02) ^a^	7.21 (±0.03) ^a^	6.22 (±0.06) ^a^	6.11 (±0.04) ^a^	Nd
**T4**	Y12	100%-Lp13	0.10 (±0.00) ^a^	0.58 (±0.05) ^b^	6.29 (±0.00) ^a^	4.38 (±0.09) ^b^	0.15 (±0.00) ^a^	0.14 (±0.00) ^a^	6.62 (±0.06) ^a^	7.80 (±0.10) ^a^	6.23 (±0.01) ^a^	6.01 (±0.17) ^a^	Nd
**T5**	Y12+Lp13+LPG1+Lpl15	100%-Lp13	0.11 (±0.01) ^a^	0.76 (±0.03) ^a^	6.33 (±0.02) ^b^	4.13 (±0.01) ^a,b^	0.15 (±0.01) ^a^	0.15 (±0.01) ^a^	6.77 (±0.09) ^a^	7.50 (±0.17) ^a^	6.15 (±0.08) ^a^	6.04 (±0.06) ^a^	Nd
**T6**	Control	100%-Lp13	0.12 (±0.06) ^a^	0.81 (±0.01) ^a^	6.22 (±0.05) ^a^	4.18 (±0.06) ^a,b^	0.12 (±0.06) ^a^	0.15 (±0.00) ^a^	6.63 (±0.15) ^a^	7.43 (±0.31) ^a^	6.94 (±0.07) ^b^	5.92 (±0.04) ^a^	Nd

* TA. Titratable Acidity, ** CA. Combined Acidity, Nd. Not detected. LAB were absent before inoculation. Different superscript letter, within the same column, are significantly different (*p* ≤ 0.05) according to post-hoc comparison test.

**Table 2 foods-08-00280-t002:** Contribution of treatments (inoculation with different LAB and yeast species) to the production of the different VOCs found in brine at the end of the fermentation as assessed by their standardized effects. Only 71 significant compounds (from a total of 131) with differences between at least two treatments were used for these analyses. Contribution of treatments for the different VOCs was compared with respect to the spontaneous fermentation (T6). T1 stands for treatment inoculated with LPG1, T2 with Lp13, T3 with Lpl15, T4 with Y12, and T5 with Y12+LPG1+Lp13+Lpl15.

		Contributor (ANOVA) and Sign (Standaridised Coefficient)						PLS-R Analysis. Significant Coefficients		
**Compound**	Code	T1	T2	T3	T4	T5	Pooled SD	TA *	pH	CA **
**Methyl acetate**	A	-	-	-	0.710	0.703	0.155	−0.351 (±0.092)	-	−0.171 (±0.046)
**Ethyl acetate**	B	-	-	-	0.957	0.470	0.053	−0.464 (±0.199)	0.345 (±0.114)	−0.226 (±0.104)
***cis*-3-Hexenyl acetate**	C	-	−0.542	-	-	-	0.226	−0.187 (±0.061)	-	−0.091 (±0.028)
**2-Phenylethyl acetate**	D	0.596	-	-	0.970	0.625	0.024	-	-	-
**3-Methylbutanoic acid**	E	-	-	-	-	0.725	0.185	-	-	-
**Methanol**	F	−0.341	-	0.771	−0.242	-	0.112	-	-	-
**Ethanol**	G	-	-	-	0.750	-	0.147	−0.459 (±0.144)	0.341 (±0.135)	−0.223 (±0.072)
**2-Butanol**	H	0.688	-	-	0.377	0.764	0.132	-	-	-
**2-Methyl-1-propanol**	I	-	−0.819	-	-	−0.819	0.152	-	-	-
**1-Butanol**	J	0.501	0.413	0.351	1.208	0.753	0.048	-	0.308 (±0.103)	-
**2-Methyl-1-butanol**	K	-	−0.346	-	0.798	-	0.090	-	0.348 (±0.118)	−0.227 (±107)
**3-Methyl-1-butanol**	L	-	−0.430	-	0.738	-	0.101	−0.449 (±0.179)	0.334 (±0.121)	−0.219 (±0.088)
**3-Methyl-3-buten-1-ol**	M	0.684	−0.200	-	0.587	0.650	0.079	-	-	-
**1-Pentanol**	N	-	-	-	0.816	0.436	0.123	−0.351 (±0.159)	-	−0.171 (±0.069)
***cis*-2-Penten-1-ol**	*O*	0.774	-	-	0.760	0.765	0.046	-	-	-
**2-Methyl-2-buten-1-ol**	P	0.613	-	-	0.631	0.502	0.160	-	-	-
**1-Hexanol**	Q	0.399	-	-	0.637	0.539	0.146	-	-	-
***cis*-3-Hexen-1-ol**	*R*	0.564	−0.301	-	0.464	0.604	0.108	-	-	-
**2-Methyl-3-hexanol**	S	0.611	-	-	0.881	0.389	0.142	-	-	-
**1-Heptanol**	T	-	−0.428	-	0.744	-	0.117	−0.394 (±0.160)	+0.293 (±0.126)	−0.192 (±0.071)
**6-Hepten-1-ol**	U	-	-	-	0.749	-	0.165	−0.421 (±0.164)	+0.313 (±0.133)	−0.205 (±0.081)
***cis*-5-Octen-1-ol**	*V*	-	−0.385	-	0.615	-	0.178	−0.376 (±0.132)	+0.280 (±0.120)	−0.183 (±0.062)
**Benzyl alcohol**	W	0.283	−0.282	-	0.788	0.433	0.090	-	+0.237 (±0.099)	-
**2-Phenylethanol**	X	0.558	−0.339	−0.122	0.665	0.466	0.038	-	-	-
**2-Ethenyl-2-butenal**	Y	-	-	0.370	−0.412	−0.412	0.147	-	−0.221 (±0.064)	-
**Isoxylaldehyde**	Z	-	-	0.670	-	-	0.212	-	-	-
**Dimethyl Sulfoxide**	AA	-	-	-	-	-	0.212	-	-	-
**β-Damascenone**	AB	−0.667	-	-	-	-	0.201	−0.348 (±0.147)	-	−0.169 (±0.068)
**Ethyl lactate**	AC	-	-	-	0.470	-	0.191	-	-	−0.111 (±0.045)
**Ethyl 5,6-dimethylnicotinate**	AD	-	-	0.129	0.981	-	0.045	-	+0.369 (±0.122)	-
**Unknown ester (m/z 88)**	AE	−0.732	-	-	−0.767	-	0.178	-	-	-
**Furfuryl methyl ether**	AF	-	−0.349	-	-	−1.031	0.103	-	-	-
**Acetoin**	AG	−0.358	-	−0.328	−0.319	0.567	0.143	-	-	-
**6-Methyl-3,5-heptadien-2-one**	AH	-	-	0.363	0.495	-	0.183	−0.344 (±0.120)	+0.256 (±0.084)	−0.167 (±0.066)
**Purpurocatecho**	AI	-0.151	−0.251	-	−0.887	−0.887	0.065	-	-0.259 (±0.077)	-
**Iridomyrmecine**	AJ	-	−0.496	-	0.324	−0.688	0.118	-	-	-
**Methyl lactate**	AK	-	-	-	−0.864	-	0.132	-	-0.328 (±0.108)	-
**Methyl hydrocinnamate**	AL	-	-	-	−0.673	-	0.217	-	-	-
**Methyl 4(methylamino)benzoate**	AM	-	-	-	0.786	-	0.136	−0.373 (±0.142)	+0.278 (±0.108)	−0.182 (±0.076)
**3-Ethylpyridine**	AN	−0.457	-	-	-	−0.526	0.182	-	-	-
**4-Methylguaiacol**	AO	0.347	−0.483	-	0.662	-	0.100	-	-	-
**4-Ethylguiacol**	AP	0.705	−0.263	−0.196	0.608	0.216	0.067	-	-	-
**4-Ethylphenol**	AP	-	-	0.632	−0.407	-	0.149	-	-	-
**Isovanillic acid**	AR	−0.239	−0.551	-	0.260	−0.739	0.111	-	+0.184 (±0.081)	-
**Coumaran**	AS	−1.093	−0.898	−0.571	−7.460	−0.903	0.109	-	-	-
**5-tert-Butylpyrogallol**	AT	-	−0.610	−0.455	0.781	−0.371	0.097	-	-	-
**Methoxyeugenol**	AU	−0.247	−0.430	-	0.366	−0.723	0.105	-	+0.220 (±0.092)	-
**Vainillin**	AV	-	-	−0.692	−0.799	−0.529	0.187	+0.320 (±0.075)	-	+0.156 (±0.036)
**α-Isophorone**	AW	−0.481	−0.584	-	-	−0.513	0.200	-	-	-
**α-Terpineol**	AX	-	-	-	-	0.572	0.198	-	-	-
**Geraniol**	AY	0.712	−0.297	-	-	0.546	0.127	-	-	-
**Unknown A (m/z 71-59)**	AZ	-	-	-	-	−0.627	0.187	-	-	-
**Unknown B (m/z 123-138-96)**	BA	-	−0.402	-	0.590	-	0.167	-	+0.279 (±0.107)	-
**Unknown C (m/z 83-112-97)**	BB	0.270	-	-	0.974	-	0.098	-	+0.300 (±0.137)	-
**Unknown D (m/z 55-93-108)**	BC	0.151	-	-	1.005	0.506	0.037	-	+0.329 (±0.116)	-
**Unknown E (m/z 111-198)**	BD	-	−0.326	-	0.808	-	0.125	-	+0.309 (±0.115)	-
**Unknown F (m/z 95-154-110)**	BE	0.261	-	-	0.940	0.665	0.085	−0.376 (±0.159)	−0.183 (±0.072)	-
**Unknown G (m/z 138)**	BF	−0.463	−0.664	-	-	−0.716	0.153	-0.250 (±0.106)	−0.121 (±0.051)	-
**Unknown H (m/z 113-81-153)**	BG	0.548	−0.457	-	0.518	-	0.127	-	-	-
**Unknown I (m/z 99-139-67-81)**	BH	-	−0.413	-	−0.807	-	0.179	-	−0.296 (±0.129)	-
**Unknown K (m/z 93-79)**	BI	-	−0.414	0.289	-	−0.754	0.123	-	-	-
**Unknown L (m/z 222-43-85-177)**	BJ	−0.822	−0.612	-	−0.576	−0.776	0.218	-	-	-
**Unknown M (m/z 138-120)**	BK	−0.629	−0.799	-	-	−0.710	0.186	-	-	-
**Unknown N (m/z 151-43)**	BL	-	-	-	−0.894	-	0.157	-	−0.344 (±0.113)	-
**Unknown O (m/z 95-110-138)**	BM	−0.704	−0.859	-	−0.721	−0.370	0.195	-	-	-
**Unknown P (m/z 138)**	BN	−0.669	−0.644	-	-	−0.694	0.172	-	-	-
**Unknown Q (m/z 102-55-69)**	BO	−0.208	−0.194	-	0.866	0.235	0.059	−0.497 (±0.185)	+0.70 (±0.130)	−0.242 (±0.097)
**Unknown S (m/z 167-121)**	BP	0.364	−0.613	-	-	-	0.157	-	-	-
**Unknown T (m/z 70-55-82)**	BQ	-	-	-	−0.590	-	0.217	-	−0.245 (±0.065)	-
**Unknown U (m/z 119-159-192)**	BR	−0.457	-	-	−0.928	−0.528	0.181	-	-	-
**Unknown W (m/z 121-136-161)**	BS	-	−0.761	-	-	-	0.193	-	-	-

Notes: LAB and yeast columns were removed from the PLS-R information since these variables were never significant. * TA, titratable acidity; ** CA, combined acidity. In parenthesis standard errors.

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
