# Peer review of "Lactic Acid Bacteria and Yeast Inocula Modulate the Volatile Profile of Spanish-Style Green Table Olive Fermentations"

_foods, 2019, doi:10.3390/foods8080280_

Round 1
Reviewer 1 Report
The work combined microbiological (lactic acid bacteria and yeasts), physicochemical (pH, acidity, combined acidity) and volatile compounds (VOCs) as derived by GC/MS in tandem with multivariate data analysis to infer on the effect of inoculated fermentation of Spanish-style green olives using selected cultures of LAB and yeasts. Overall, the work is very interesting and the data collected are analyzed in-depth using a variety of data analysis methods. To begin with, I have a comment of a general nature to make. The analysis of VOCs was undertaken at day 65 (about 2 months) that was considered the end of fermentation. However, volatile compounds are a dynamic system that changes during time. So, this profile of compounds is a snapshot at this particular time point. Will it be the same after 3 months of storage in the fermentation vessels? In addition, the profile of VOCs was associated with the selected cultures. This means that the selected starter cultures dominate the process in order to be able to affect the profile of volatile compounds. So, it is necessary to make molecular analysis to define the survival of selected cultures at the end of the process. This was done by the authors for the inoculated LAB cultures but not for the yeast Y12 used in fermentation. First, I would like to ask if the selection of 10 colonies from each treatment (I guess it is from the highest dilution) is enough for a representative analysis of the dominant population. Nevertheless, this was not done for the yeast starter. Could the authors provide a justification on this issue? What was the dominance percentage of Y12 either alone or in the mixed culture.
Line 113: Please indicate in this paragraph how many times you analyzed each sample.
Table 2, page 9: Please indicate the meaning of the column defined as “Reference”.
Page 10, last 3 lines (there is no line numbering in the manuscript): the authors indicate that the effect of inoculation treatment on the volatile profile is presented by a + sign meaning increase versus the spontaneous process (control) and a – sign meaning decrease. However, I cannot see any + sign on table 2, especially for 3-methyl-1-butanol as the authors point out.
Figures 4A and B are not clear. Please increase the quality of the figures.
Figure 5 presents a biclustering which is an unsupervised method that already includes the information presented in Figure 3. So, the two figures provide the same information as far as the clustering of the treatments is concerned. As Figure 5 is more informative about the process, I propose to the authors to keep only Figure 5 in their paper.
Author Response
We have revised and modified it according to your suggestions and comments. All the suggestions have been very helpful to improve the manuscript.
Question (Q1). The analysis of VOCs was undertaken at day 65 (about 2 months) that was considered the end of fermentation. However, volatile compounds are a dynamic system that changes during time. So, this profile of compounds is a snapshot at this particular time point. Will it be the same after 3 months of storage in the fermentation vessels?
Answer (A1). Effectively, the profile of the volatile compounds is a dynamic system. However, due to the fermentation conditions, a progressive accumulation of the successive formed compounds is also produced. In this way, the profile at the end of fermentation it is expected to represent a compendium of those produced throughout the whole process. Some comments on this aspect have been introduced in the revised version of the manuscript. Besides, similar methodology has been already used by other authors like Randazzo et al. 2017 (Randazzo, C.L.; Todaro, A.; Pino, A.; Pitino, I.; Corona, O.; Caggia, C. Microbiota and metabolome during controlled and spontaneous fermentation of Nocellara Etnea table olives. Food Microbiol. 2017, 65, 136–148) who reported that the highest differences in volatile compound profiles produced by diverse microorganisms was found at the end of fermentation.
Q2. In addition, the profile of VOCs was associated with the selected cultures. This means that the selected starter cultures dominate the process in order to be able to affect the profile of volatile compounds. So, it is necessary to make molecular analysis to define the survival of selected cultures at the end of the process. This was done by the authors for the inoculated LAB cultures but not for the yeast Y12 used in fermentation. First, I would like to ask if the selection of 10 colonies from each treatment (I guess it is from the highest dilution) is enough for a representative analysis of the dominant population. Nevertheless, this was not done for the yeast starter. Could the authors provide a justification on this issue? What was the dominance percentage of Y12 either alone or in the mixed culture.
A2. The work hypothesis of this study was to prove that the VOC profile of olive fermentation could be modulate by addition of starter culture, as effectively corroborated. For this reason, only the usual routine inoculum imposition through fermentation was followed, especially for LAB at the moment of maximum population. After the valuable conclusion obtained in this initial work, exclusively focused on the possible modulation on the volatile profiles by the starters used, further studies clearly should include a more detailed research on the diverse groups of microorganisms involved. Additional information of the dilution where LAB isolates were obtained has been introduced in the revised version of the manuscript.
Q3. Line 113: Please indicate in this paragraph how many times you analyzed each sample.
A3. Samples (fermentation vessel) were analyzed individually, to obtain average by treatment. This information was introduced in the revised version of the manuscript.
Q4. Table 2, page 9: Please indicate the meaning of the column defined as “Reference”.
A4. The meaning was Code. The word “Reference” has been substituted with “Code”
Q5. Page 10, last 3 lines (there is no line numbering in the manuscript): the authors indicate that the effect of inoculation treatment on the volatile profile is presented by a + sign meaning increase versus the spontaneous process (control) and a – sign meaning decrease. However, I cannot see any + sign on table 2, especially for 3-methyl-1-butanol as the authors point out.
A5. The absence of sign was thought to be interpreted as positive. To prevent any misunderstanding, the sign + was introduced in Table 2.
Q6. Figures 4A and B are not clear. Please increase the quality of the figures.
A6. The only way of doing it has been by removing color from the new figure.
Q7. Figure 5 presents a biclustering which is an unsupervised method that already includes the information presented in Figure 3. So, the two figures provide the same information as far as the clustering of the treatments is concerned. As Figure 5 is more informative about the process, I propose to the authors to keep only Figure 5 in their paper
A7. As suggested by reviewer, Figure 3 was removed from the revised manuscript and their comments conveniently integrated into biclustering, which is a new subsection now.
Reviewer 2 Report
Lactic acid bacteria and yeast inocula modulate the volatile profile of Spanish-style green table olive fermentations
Manuscript Foods_554874
#REVIEWER
Line 113: Olive brines’ sequential extraction and GC-MS analysis
According to some authors (Tufariello et al “"Biotechnology can improve a traditional products as table olives" chapter of book 2016), the treatment with NaOHcauses the disruption of many aroma compounds together with nutritional and health important molecules, therefore it would have been useful to characterize the volatile fraction of the table olives before and after different treatments carried out by different yeasts and or bacteria strains. Why the authors have omitted table olives volatile characterization?
Results
I suggest reviewing the structure of the results which appears more substantial than the discussions. Some parts reported in the results are discussions
Notes:
1. Figure 4 is not clear,it is asked to improve it;
2. The lines are not shown on the whole manuscript.

Author Response
We have revised and modified it according to your suggestions and comments. All the suggestions have been very helpful to improve the manuscript.
Question (Q1). Line 113: Olive brines’ sequential extraction and GC-MS analysis. According to some authors (Tufariello et al “"Biotechnology can improve a traditional products as table olives" chapter of book 2016), the treatment with NaOH causes the disruption of many aroma compounds together with nutritional and health important molecules, therefore it would have been useful to characterize the volatile fraction of the table olives before and after different treatments carried out by different yeasts and or bacteria strains. Why the authors have omitted table olives volatile characterization?
Answer (A1). In this case, all olives were subjected to similar NaOH treatment, thus eliminating the effect of such operation in the design.
Q2. Why the authors have omitted table olives volatile characterization?
A2. Thank you to reviewer for his/her comment. However, the table olive volatile characterization was outside the scope of the work. Its characterization was not necessary since the NaOH treatment effect, as commented previously, was removed from the design by applying it similarly to all olive fermentations.
Q3. Results. I suggest reviewing the structure of the results which appears more substantial than the discussions. Some parts reported in the results are discussions
A3. Thank you reviewer for his/her comment. We have reviewed in detail both results and discussion section, but after diverse reading and consideration, we have kept in its original version.
Q4. Notes: 1. Figure 4 is not clear, it is asked to improve it;
A4. Figure 4 (Fig. 3 in the revised manuscript) has been replaced for a new one, which we consider clearer. Notice that, in this way, the presentation of results also was improved.
Q5 Notes:2. The lines are not shown on the whole manuscript.
A5. We suppose that the reviewer refers to the numbering of the lines. In this case, it must be emphasize that manuscript lines were correctly numbered in the original copy sent to the Journal. The absence of line numbering in the copy provided to reviewers can then not be attributed to authors’ negligence.
Reviewer 3 Report
This article aimed to determine the contribution of lactic acid bacteria (LAB) and yeast native starter cultures on the volatile organic compound (VOC) profiles of Spanish-style green olive brines versus the spontaneous fermentation. Authors used diverse multivariate statistical techniques for the interpretation of the results. They reported that the use of mixed/sequential LAB and yeasts inocula in Spanish-style green table olive fermentation could be advisable for the industry to produce differentiated and high quality products with improving aromatic profile.
The article is straightforward. Although the article is not innovative, it contains original and interesting information. This article could have been improved if the authors clarify and/or adhere to some of the comments addressed below.
What would be the possible explanation that the level of LAB in T6 (spontaneous, fermented with naturally occurring LAB and yeast) is significantly higher than others (T1-T5), which are inoculated with LAB? Assessing ecological community of microorganisms (indigenous microbiota) in the sample systems before and after fermentation may manifest possible interaction of indigenous microbiota resulting in this phenomena. Are all of these analysis methods necessary? Some of analysis methods seem to reveal similar results as other methods.
Line 111. Revise to “… picked for their fingerprinting and compared with …”.
Line 114. Revise to “… treatment stored at…”.
Lines 194-197. Authors implied that each strain of LAB affects VOC profile. Authors reported that Lp13 was detected in 100% frequency in the treatments of T4-T6. If so, what level of Lp13 was detected in the treatments? Additionally, was there any Lp13 in the treatments of T1-T3 at all? This question arises since control sample (T6) had Lp13. Although the level might not be significant, Lp13 might have been present in treatments (T1-T3) as well.
Lines 201-203. Statements related to statistical analysis may need to be revised. Statistical difference between “a” and “ab” is not significant.
Table 1. Authors reported high levels of LAB and yeast in T6. What strains are they consisted of? Initial microbial population of samples may be needed. Authors may insert “±“prior to numbers in parenthesis [i.e., (±0.06)].
Page 11. Revise to “…process (T6) with only a few coefficients with negative sign.”
Page 11. Revise to “…associated with Lp13 presence, which did not promote their production.”
Author Response
We have revised and modified it according to your suggestions and comments. All the suggestions have been very helpful to improve the manuscript.
Question (Q1). What would be the possible explanation that the level of LAB in T6 (spontaneous, fermented with naturally occurring LAB and yeast) is significantly higher than others (T1-T5), which are inoculated with LAB? Assessing ecological community of microorganisms (indigenous microbiota) in the sample systems before and after fermentation may manifest possible interaction of indigenous microbiota resulting in this phenomena. Are all of these analysis methods necessary? Some of analysis methods seem to reveal similar results as other methods.
Answer (A1).The evolution of the LAB populations in the treatments could be considered as normal. Usually, during the first days of fermentation, the LAB populations in the inoculated treatments overpass those in the spontaneous process. This produces a more rapid acidification and depletion of nutrients in the inoculated treatments, with the earlier subsequent decrease in the LAB population as time progresses. However, in case of the spontaneous process, the evolution is slower and the LAB population takes more time to reach its maximum and initiate its decline. Since the LAB counts were obtained after 65 days of fermentation, it can be normal that the LAB population in the spontaneous process were still higher at this moment. This point was clarified in the revised version of manuscript.
Q2. Line 111. Revise to “… picked for their fingerprinting and compared with …”.
A2. The correction was introduced in the revised version of the manuscript.
Q3. Line 114. Revise to “… treatment stored at…”.
A3. The suggestion was introduced in the revised version of the manuscript.
Q4. Lines 194-197. Authors implied that each strain of LAB affects VOC profile. Authors reported that Lp13 was detected in 100% frequency in the treatments of T4-T6. If so, what level of Lp13 was detected in the treatments? Additionally, was there any Lp13 in the treatments of T1-T3 at all? This question arises since control sample (T6) had Lp13. Although the level might not be significant, Lp13 might have been present in treatments (T1-T3) as well.
A4. As previously mentioned, a detailed control of the imposition, inoculum performance or LAB populations was outside the scope of the work, which was mainly focused on the effect that the addition of inoculum could have on VOC profiles.
Q5. Lines 201-203. Statements related to statistical analysis may need to be revised. Statistical difference between “a” and “ab” is not significant.
A5. The idea was just to emphasize high values. Nevertheless, the sentence was corrected to prevent any misunderstanding.
Q6. Table 1. Authors reported high levels of LAB and yeast in T6. What strains are they consisted of? Initial microbial population of samples may be needed. Authors may insert “±“prior to numbers in parenthesis [i.e., (±0.06)].
A6. This LAB population obtained in T6 treatment was determined at the end of fermentation, as it was mentioned previously. The LAB population level was also measured before inoculation without detecting them in any treatment. This information was introduced in the revised version of manuscript. Besides the ± sign was also inserted.
Q7. Page 11. Revise to “…process (T6) with only a few coefficients with negative sign.”
A7. The typo error was corrected.
Q8. Page 11. Revise to “…associated with Lp13 presence, which did not promote their production.”
A8. The suggestion was introduced in the revised version of the manuscript.
Reviewer 4 Report
o far my comments on the article " Lactic acid bacteria and yeast inocula modulate the volatile profile of Spanish-style green table olive fermentations":
L52-53: ...should read "during the olive fermentation were significantly affected..."?
L114: ....and stored at... ?
In many cases, it is written 'physic-chemical' parameters instead of 'physico-chemical'?
L199-200: The sentence can be better revised for clarity
Unfortunately, the line numbering stopped on page 6,
On
page 10: Please check the sentence - Should it be "This indicates that
the presence of the LAB did not interfere with its possible formation
but even stimulated it.."?
Under Discussion:
It will be interesting for readers to get a better grasp of why the presence of Y12 dramatically increased the concentration of most of the VOCs over the T6 process, with more reference to previous similar studies. Similarly, refer to more studies on yeast inoculation with LAB strains (T5).
Author Response
We have revised and modified it according to your suggestions and comments. All the suggestions have been very helpful to improve the manuscript.
Question (Q1). L52-53: ...should read "during the olive fermentation were significantly affected..."?
Answer (A1). The change was introduced.
Q2. L114: ....and stored at... ?
A2. The suggested change was incorporated in the revised version of the manuscript.
Q3. In many cases, it is written 'physic-chemical' parameters instead of 'physico-chemical'?
A3. There is no a complete agreement on the use of physico-chemical or physicochemical terms. Apparently, the second form is gaining popularity. So, according to reviewer’s suggestion, it was adopted thought the revised version of the manuscript.
Q4. L199-200: The sentence can be better revised for clarity
A4. The sentence was removed from the revised version of the manuscript.
Q5. On page 10: Please check the sentence - Should it be "This indicates that the presence of the LAB did not interfere with its possible formation but even stimulated it.."?
A5. The sentence was modified according to reviewer’s suggestion.
Q6. It will be interesting for readers to get a better grasp of why the presence of Y12 dramatically increased the concentration of most of the VOCs over the T6 process, with more reference to previous similar studies. Similarly, refer to more studies on yeast inoculation with LAB strains (T5).
A6. To obtain information about studies of the use of mixed starter LAB+yeast in table olives, a comprehensive bibliographic search in Scopus and PubMed was performed and the references introduced in the original version of the paper. However, any further contributions which could be missing would be welcome.